# Carbon sequestration during core formation implied by complex carbon polymerization

Natalia V. Solomatova [1], Razvan Caracas [1] & Craig E. Manning[2]

Current estimates of the carbon flux between the surface and mantle are highly variable, and the total amount of carbon stored in closed hidden reservoirs is unknown. Understanding the forms in which carbon existed in the molten early Earth is a critical step towards quantifying the carbon budget of Earth's deep interior. Here we employ first-principles molecular dynamics to study the evolution of carbon species as a function of pressure in a pyrolite melt. We find that with increasing pressure, the abundance of $CO_2$ and $CO_3$ species decreases at the expense of $CO_4$ and complex oxo-carbon polymers ($C_xO_y$) displaying multiple C-C bonds. We anticipate that polymerized oxo-carbon species were a significant reservoir for carbon in the terrestrial magma ocean. The presence of Fe-C clusters suggests that upon segregation, Fe-rich metal may partition a significant fraction of carbon from the silicate liquid, leading to carbon transport into the Earth's core.

[1] Laboratoire de Géologie de Lyon, CNRS UMR 5276, Université Claude Bernard Lyon 1, École Normale Supérieure de Lyon, Lyon 69007, France.
[2] Department of Earth, Planetary and Space Sciences, University of California, Los Angeles 90095 CA, USA. Correspondence and requests for materials should be addressed to N.V.S. (email: nsolomat@gmail.com)

I t is critical to characterize the distribution and storage mechanisms of carbon in the deep Earth to improve our understanding of the global carbon cycle. Presently, the carbon budget of the deep Earth is not well constrained and is highly model-dependent[1–4]. Estimates of the amount of carbon exchanged between the surface and mantle range between 30 and 130 megatons per year[1,5] while estimates of the carbon concentration within the core range between 0.2 and 4 wt%[2,6,7]. In silicate melts of the uppermost mantle, carbon exists predominantly as $CO_2$ and $CO_3$[8–11]; however, at greater depths, the speciation of carbon has not been investigated in sufficient detail, because the form in which carbon is dissolved in silicate melts and the mechanism by which carbon polymerizes (where carbon atoms bond to other carbon atoms) is difficult to determine experimentally. The majority of experiments require quenching of the melt prior to analysis, which may change the local structure of carbon[12]. Conversely, ab initio calculations allow one to determine the precise behavior of atoms at the high temperatures and pressures of planetary interiors and directly calculate thermodynamic properties.

The speciation of carbon at high pressure has direct consequences for the storage mechanism and distribution of carbon in Earth's interior. Whether the majority of deep carbon is stored in the core or in the lower mantle as diamonds, carbides, or carbonates is affected by the evolution and fate of carbon during the magma ocean phase of Earth. It has recently been shown that diamonds can form from a reaction of carboxylates with water at constant oxygen fugacity[13], providing an unexplored mechanism for diamond genesis. At ambient pressure, carbon behaves as a network modifier (i.e., exists as $CO_x$ species that break up the silicate network) while at high pressure it is thought that carbon will behave as a network former by replacing silicon. However, it has been shown that carbon atoms may bond directly to silicon in a highly reduced carbon-rich melt, suggesting that carbon in silicate melts may resemble oxygen more than silicon at high pressures[14].

The detailed computational study presented here provides insights into high-pressure, high-temperature carbon chemistry without the limitations inherent to experiments conducted at extreme conditions. We employ molecular dynamics with density functional theory to explore carbon's coordination environment in complex silicate melts relevant to Earth's early magma ocean stage. We used a pyrolite composition[15] with 4CO, 4$CO_2$, 8CO, and 8$CO_2$ units added to a 153-atom supercell, which equates respectively to 3.35 wt% CO, 5.16 wt% $CO_2$, 6.48 wt% CO, and 9.82 wt% $CO_2$, hereafter referred to as "pyr4CO", "pyr4$CO_2$", "pyr8CO", and "pyr8$CO_2$", respectively. The chosen carbon concentrations (1.4–2.8 wt% C) reflect the range of carbon contents of carbonaceous chondrites[16] (0–6 wt% C), the building blocks of Earth. It is thought that the oxidation state of Early Earth's mantle was far below the iron-wüstite buffer prior to core–mantle separation, subsequently becoming more oxidized due to convective mixing[17,18]. The addition of $CO_2$ to pyrolite melt yields oxygen abundance relevant to more oxidized magma-ocean conditions, whereas the addition of CO simulates more reduced magma-ocean conditions. We find that at 100 GPa, about half of carbon atoms are polymerized (i.e., bonded to carbon) in pyrolite melts with 3.35–6.48 wt% CO with significant iron–carbon clustering. The more reduced pyrolite melts yield more carbon–carbon and carbon–iron bonding.

## Results

**Equation of state**. The effect of carbon on the melt's compressibility behavior is consistent with previous work[19–21] (Supplementary Table 1). We show that the density difference between

carbon-bearing pyrolite melt and carbon-free pyrolite melt decreases with increasing pressure, reaching a local minimum at ~30 GPa after which the density difference between the melts increases again (Supplementary Fig. 1). The larger density difference at low pressure (<30 GPa) compared to high pressure (>30 GPa) suggests that carbon-bearing melts will be less buoyant at lower-mantle pressures than at pressures of the upper mantle (see Supplementary Note 1 for more details).

**Carbon coordinated by oxygen**. To discern the behavior of oxidized carbon in the magma ocean, we examine the relative proportions of $CO_2$, $CO_3$, and $CO_4$ species in pyr4$CO_2$ and pyr8$CO_2$ as a function of pressure at 4000 K (see Supplementary Fig. 2 for the relative proportions of all carbon species). Between 0 and 10 GPa, the $CO_2$ concentration decreases sharply from about 40 and 50% for pyr4$CO_2$ and pyr8$CO_2$, respectively, to about 20%, followed by a more gradual decrease in concentration to 0–15% at lowermost mantle pressures irrespective of the composition (Fig. 1a). In both pyr4$CO_2$ and pyr8$CO_2$ systems, there is a sharp increase in $CO_3$ species in the 0–10 GPa range, after which the concentration decreases with increasing pressure to about 35% at 100 GPa. Concurrently, the concentration of $CO_4$ increases sharply with increasing pressure, reaching 60% at 100 GPa for both pyr4$CO_2$ and pyr8$CO_2$ systems. The shift in carbon coordination from predominantly three-fold to four-fold is reminiscent of carbon behavior in crystalline carbonates. At ambient conditions, natural carbonates contain three-fold carbon; however, in high-pressure polymorphs of $CaCO_3$ and $MgCO_3$, carbon increases its coordination to four-fold at 75 and 83 GPa, respectively[22,23] while in $Ca_3CO_5$, a newly-proposed theoretical carbonate polymorph, fourfold carbon appears at pressures as low as 11 GPa[24]. Thus, the pressure range of the transition from $CO_3$ to $CO_4$ in silicate melts encompasses the predicted or observed coordination transition pressures in crystalline carbonate phases.

Compared to the oxidized pyrolite melts (pyr4$CO_2$ and pyr8$CO_2$), the reduced pyrolite melts (pyr4CO and pyr8CO) have a higher concentration of molecular $CO_2$ species and a lower concentration of $CO_3$ and $CO_4$ at all pressures (Supplementary Fig. 2). In other words, the coordination of carbon by oxygen is higher when more oxygen is available in the system. Comparing our results to $MgSiO_3$ enstatite melt with 5.2 and 16.1 wt% $CO_2$[21] ("en4$CO_2$" and "en14$CO_2$", respectively), we find an excellent agreement in relative proportions of $CO_2$, $CO_3$, and $CO_4$ species despite the different bulk chemistries. Thus, we find that the relative proportions of $CO_x$ are most strongly dependent on the pressure and oxidation state rather than a difference in bulk chemistry. Guillot and Sator[25] conducted molecular dynamics simulations on volcanic melts at 2–15 GPa and 1473–2273 K, also finding that the ratio of $CO_2$-to-$CO_3$ is negatively correlated to pressure and positively correlated to temperature. There have been several infra-red (IR) spectroscopy studies on quenched sodium-aluminum silicates, diopside and olivine melilite melts suggesting that the ratio of $CO_2$ to $CO_3$ is positively correlated to the silica content at pressures of 1–4 GPa and temperatures of 1500–1900 K[9,11,26–28]. For example, Brooker et al.[11] found that carbon in $NaAlSiO_4$ (42 wt% $SiO_2$) exists almost exclusively as $CO_3$, while carbon in $NaAlSi_6O_{14}$ (81 wt% $SiO_2$) exists almost exclusively as $CO_2$. pyr4$CO_2$–pyr8$CO_2$ (~42 wt% $SiO_2$) and en4$CO_2$–en14$CO_2$ (~60 wt% $SiO_2$) contain more $CO_3$ than $CO_2$ at equivalent pressures and 4000 K. The concentration of $CO_3$ increases with decreasing temperature (Supplementary Fig. 2), in agreement with the observed trend in albitic and dacitic glasses of ref. [28], and so we anticipate that lowering the temperature to 1500–1700 K would further increase the $CO_3$ content.

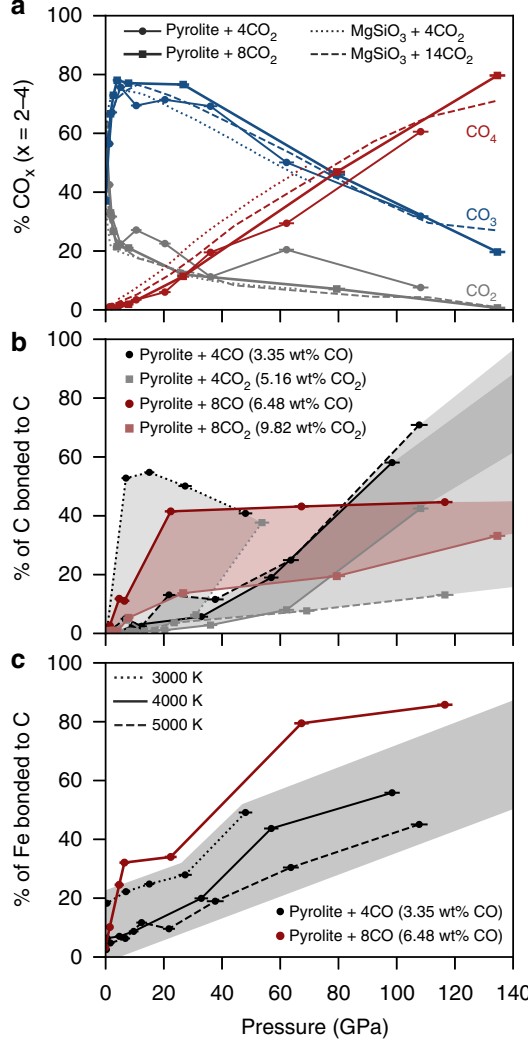

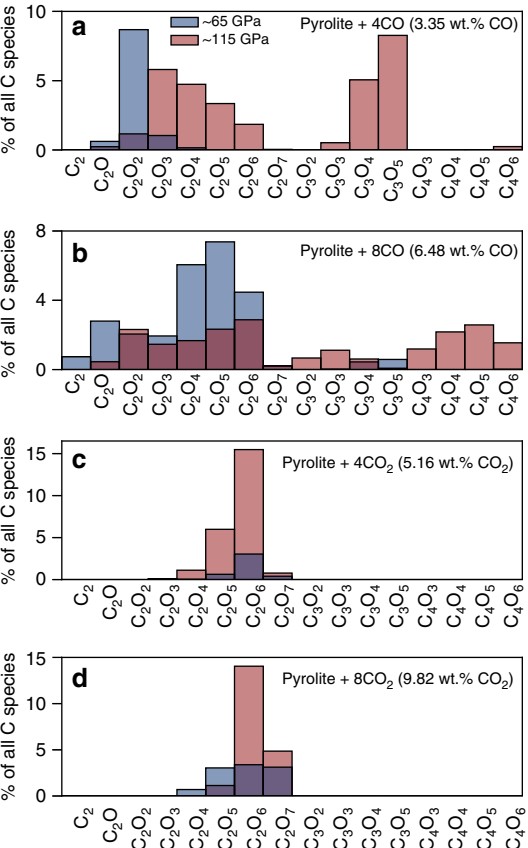

**Fig. 1** Carbon chemistry and polymerization. **a** Relative abundance of $CO_2$, $CO_3$, and $CO_4$ as a function of pressure for pyrolite + 5.16 wt% $CO_2$ (4$CO_2$ units per cell) and pyrolite + 9.82 wt% $CO_2$ (8$CO_2$ units per cell) at 4000 K. We compare our results to those of ref. [21] for $MgSiO_3$ + 5.2 wt% $CO_2$ (4$CO_2$ units per cell) and $MgSiO_3$ + 16.1 wt% $CO_2$ (14$CO_2$ units per cell), also at 4000 K. **b** The percent of polymerized carbon (i.e., the percent of carbon atoms that are bonded to another carbon atom) as a function of pressure for pyrolite + 4CO (3.35 wt% CO), pyrolite + 4$CO_2$ (5.16 wt% $CO_2$), pyrolite + 8CO (6.48 wt% CO), and pyrolite + 8$CO_2$ (9.82 wt% $CO_2$) at 3000 K (dotted line), 4000 K (solid line), and 5000 K (dashed line). Shaded regions represent the range from the more oxidized to the more reduced melts. for each temperature and/or carbon content. **c** The percent of iron atoms that are bonded to carbon in pyrolite + 4CO (3.35 wt% CO) and pyrolite + 8CO (6.48 wt% CO) at 3000–5000 K. Shaded region represents the approximate region that pyrolite + 4CO encompasses to highlight the contrast with pyrolite + 8CO. Error bars on pressures are standard deviations determined from the simulations

**Fig. 2** Types of polymerized carbon species present in the pyrolitic melt at 4000 K. **a** Pyrolite + 4CO (3.35 wt% CO), (**b**) pyrolite + 8CO (6.48 wt% CO), (**c**) pyrolite + 4$CO_2$ (5.16 wt% $CO_2$), and (**d**) pyrolite + 8$CO_2$ (9.82 wt% $CO_2$) at average pressures of ~65 GPa (blue) and ~115 GPa (red). The y-axis is expressed as the percent of all carbon–oxygen species present in the melt, including unpolymerized carbon species. Please note the different y-axis scales

**Polymerized carbon**. At 0 GPa and 3000–5000 K, carbon exists exclusively in unpolymerized forms (e.g., CO, $CO_2$, and $CO_3$) while at higher pressures, carbon becomes increasingly polymerized (i.e., higher fraction of carbon atoms bonded to another carbon atom). We find that the more reduced pyrolite melts have a larger quantity of polymerized carbon with respect to the more oxidized pyrolite melts with similar carbon concentrations (Fig. 1b). At 4000 K, pyr8CO experiences a sharp increase in

polymerized carbon content between 0 and 20 GPa, plateauing at 40% up to megabar pressures and pyr8$CO_2$ experiences a shallow increase in polymerization to about 30% at maximum pressure. The behavior of pyr4CO at 4000 K is notably distinct with the fraction of polymerized carbon remaining below 20% up to about 50 GPa and then increasing rapidly to 60% at 100 GPa, while pyr4$CO_2$ experiencing a similar behavior with 1.5–3 times less polymerization. The degree of polymerization at 5000 K is comparable to the degree of polymerization at 4000 K; however, at 3000 K, pyr4CO experiences a much more rapid increase in polymerization at low pressures, followed by a gentle decrease in polymerization, likely due to the effect of the low temperature and high pressure on carbon's mobility (and thus, the probability of encountering another carbon atom).

We explore the types of $C_xO_y$ species present in pyrolitic melt as a function of pressure and composition in Fig. 2. With increasing pressure, the quantity of polymerized carbon species and the size/complexity of the $C_xO_y$ molecules increases. For example, in pyr4CO, most of the polymerized carbon exists in $C_2O$, $C_2O_2$, and $C_2O_3$ molecules at ~65 GPa, whereas at 115 GPa, carbon exists predominantly as $C_2O_{2-6}$ and $C_3O_{4-6}$ molecules. The oxidation state has a significant effect on the types of species present. The more oxidized pyrolitic melts only contain $C_2O_4$, $C_2O_5$, $C_2O_6$, and $C_2O_7$, with nearly no $C_3O_x$ species forming while the more reduced melts have a wide range of $C_xO_y$ species

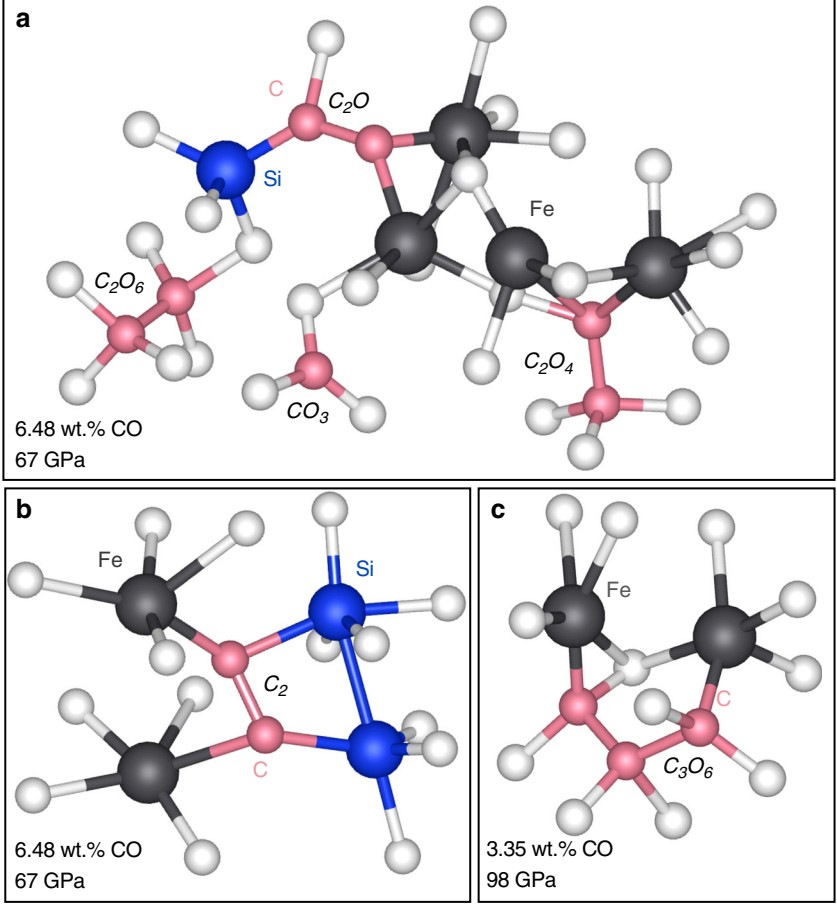

**Fig. 3** Examples of carbon bonding to iron and silicon in pyrolite melt at 4000 K. **a, b** Pyrolite + 8CO (6.48 wt% CO) at 67 GPa and **c** pyrolite + 4CO (3.35 wt% CO). The carbon species that each carbon cluster represents is indicated in italics. Atoms that are not directly bonded to the cations shown here have been omitted for clarity

present. Due to the combined effect of a low carbon concentration and low pressure, pyr4CO and pyr4CO₂ contain almost no polymerized carbon at low pressure (~6 GPa).

Additionally, we explored the effect of temperature on the polymerization of carbon, which we find is inversely dependent on temperature. For example, at 5000 K and 20 GPa, pyr4CO only has ~3% of its carbon polymerized while at 3000 K about 45% of its carbon is polymerized (Supplementary Fig. 3). Carbon polymerization in a more oxidized melt is suppressed with respect to the reduced melt at all pressures and temperatures investigated, and the effect generally increases with decreasing temperature. At approximately 20 GPa, polymerized carbon mostly exists as $C_2$, $C_2O$, and $C_2O_2$, but its concentration relative to all carbon decreases by almost one order of magnitude between 5000 and 3000 K.

**Carbon complexes with Fe and Si**. We observe strong clustering between carbon and iron, which increases with increasing pressure (Fig. 1c). At 0 GPa, less than 20% of iron is bonded to carbon for all compositions, while at 100 GPa, almost half of the iron is bonded to carbon in pyr4CO and more than 80% of iron is bonded to carbon in pyr8CO. The bonding of iron to carbon also increases with decreasing temperature; for example, at 50 GPa the percent of iron atoms bonded to carbon increases from 20 to 50% when the temperature is decreased from 5000 to 3000 K. For the more oxidized melts, the amount of iron that is bonded to carbon is lower, typically ranging between 10 and 20% at all pressures (see Supplementary Fig. 4). Nevertheless, for all the melt

compositions, nearly all carbon clusters with three or more carbon atoms contain iron above 50 GPa.

An experimental study using nuclear magnetic resonance experiments on reduced melts observed that carbon atoms bond directly to silicon in place of oxygen, suggesting that carbon may behave more like oxygen rather than silicon in reduced high-pressure silicate melts[14]. Our results confirm that carbon does in fact bond directly to silicon (Fig. 3). At 0 GPa, negligible carbon bonds to silicon; however, with increasing pressure, the fraction of carbon atoms directly bonded to silicon increases to ~25–50% at >60 GPa in the more reduced melt and ~15–30% in the more oxidized melt (Supplementary Fig. 4). The degree of bonding between carbon and silicon seems to be largely independent of temperature and carbon concentration. The carbon–silicon and carbon–iron clusters exist for up to 1 ps. There is little to no affinity of carbon to sodium, aluminum, and calcium; in fact, the radial distribution functions for carbon with respect to sodium, calcium and aluminum do not contain a first coordination sphere at all volumes and temperatures (i.e., carbon does not form bonds to these elements).

**Elemental diffusivities**. Finally, we find that the elemental diffusion coefficients in pyrolite melt are unaffected by either the presence or the abundance of carbon (Fig. 4). We observe that the network-forming cations, silicon and aluminum, have lower diffusion coefficients compared to the network-modifying cations, sodium, calcium, iron, and magnesium. The diffusion coefficient of carbon is relatively high at low pressures, but experiences a

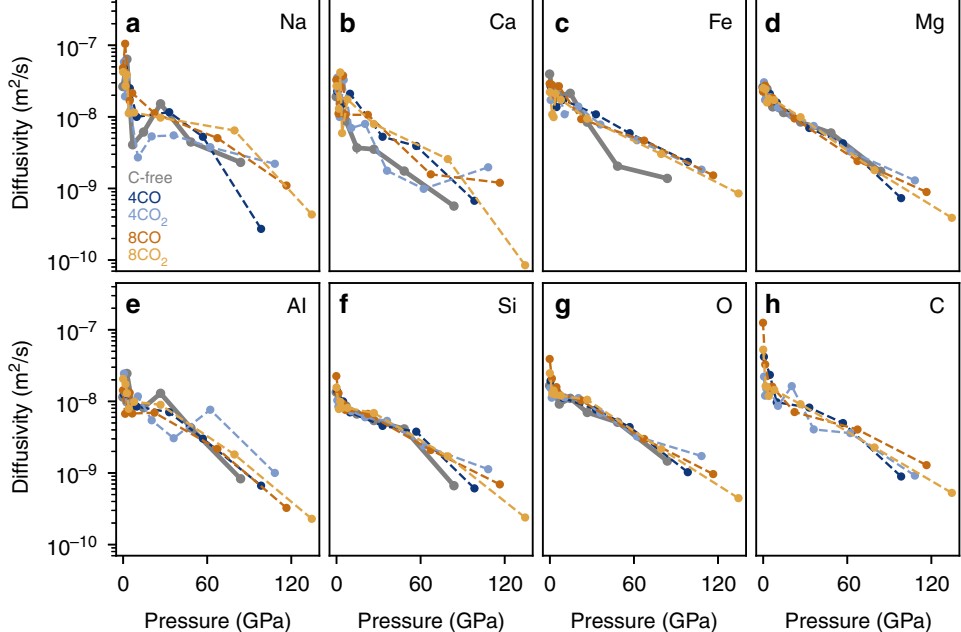

**Fig. 4** Elemental diffusivities at 4000 K. Diffusion coefficients for (**a**) sodium, (**b**) calcium, (**c**) iron, (**d**) magnesium, (**e**) aluminum, (**f**) silicon, (**g**) oxygen, and (**h**) carbon in pure pyrolite (bold gray), pyrolite + 4CO (3.35 wt% CO) (dark blue), pyrolite + 4CO$_2$ (5.16 wt% CO$_2$) (light blue), pyrolite + 8CO (6.48 wt% CO) (dark orange), and pyrolite + 8CO$_2$ (9.82 wt% CO$_2$) (yellow). The axes are identical for each panel. The scatter in the diffusion rates for Na, Ca, and Al is due to the low abundance of those atoms in our melt (our melt contains one Na, two Ca, and three Al atoms)

rapid decrease in diffusivity with increasing pressure as carbon is incorporated into the silica network. A notable result is the changing behavior of carbon: as pressure increases, carbon behaves more like a network former rather than a network modifier, because of its increased coordination by oxygen and relative long lifetimes of such clusters. This suggests that any observed or inferred differences in viscosity between the carbon-free and carbon-bearing silicate melts should come from another mechanism, not the atomic diffusion. At large depths, as carbon and silicon have similar diffusivities, and even present chemical affinity, it is not unreasonable to think that even the melt viscosity would be similar between the two types of melts.

## Discussion

The complex carbon polymerization in our simulations may represent diamond precursors. In the presence of hydrogen, oxocarbons may form carboxylates (e.g., acetate CH$_3$COO$^-$), which have been shown to form diamonds in a reaction with water[13]. Thus, our simulations reveal a possible mechanism for diamond formation in pyrolitic melts that does not necessitate the direct formation of oxygen-free carbon clusters (see Supplementary Note 2 for more details). Additional simulations are needed on pyrolite melt with carbon and hydrogen species. In any case, our results robustly reveal that oxo-carbon polymers are a previously unappreciated reservoir for carbon in high-pressure silicate liquids. The variable stoichiometry indicates that the amount of carbon stored this way flexibly adjusts to changing intensive parameters (pressure, temperature, and oxygen fugacity).

The affinity of carbon for iron implies that in the large magma oceans of the proto-Earth and Early Earth (prior to and after the Moon-forming impact) carbon would follow iron during segregation of an iron metallic liquid, and a majority of carbon present in the deep mantle may have been transported into the iron-rich core. After the moon-forming impact, metal iron and silicate melt would have been well-mixed[29] and subsequent core segregation would have resulted in siderophile elements being sequestered

into the Earth's core[30]. Although ab initio molecular dynamics simulations are not able to model phase separation, our observations of strong clustering between iron and carbon suggest that upon segregation, carbon would be sequestered into the core. Carbon is a key light-element candidate for explaining the density deficit of the core[31] and has been experimentally shown to have a high solubility in iron melt[32]; our study is the first molecular dynamics study to characterize the behavior of carbon in the presence of iron in a silicate melt.

## Methods

**Molecular dynamics simulations.** Ab initio molecular dynamics simulations based on projector-augmented wave (PAW) flavor[33] of density functional theory (DFT) were performed with the Vienna Ab Initio Simulation Package (VASP)[34] using the generalized gradient approximation (GGA) in the Perdew–Burke–Ernzerhof (PBE) form[35] to treat electron exchange and correlation. Kinetic-energy cutoffs of 550 and 800 eV were used for the plane-wave expansion of the wavefunctions and the augmentation charges, respectively. Molecular dynamics simulations were performed in the canonical ensemble (NVT) via a Nosé–Hoover thermostat[36,37] with a time step of 1–2 fs for 10–30 ps, depending on the density. The Brillouin zone sampling was performed at the gamma point. The simulations were spin-polarized at all temperatures, pressures, and compositions. The mean-square displacement as a function of time shows a ballistic regime below approximately 1000 fs, after which the atoms reach a diffusive regime (Supplementary Fig. 5).

A bond is defined by the interaction of the atoms' valence electrons while the distance between the two nuclei determines the bond length, which depends on the number of bonded electrons[38]. A pair distribution function describes the probability of finding an atom type at a given distance from the reference atom, and it allows us to determine the bond distance that includes the first coordination sphere of atoms that are directly bonded to the reference atom, becoming the bond criterion between two atom types. For example, we find that the average C–O bond length is ~1.2 Å with a distribution of approximately ±0.7 Å, resulting in a maximum bond length of about ~1.9 Å (see Supplementary Fig. 6).

We work with a pyrolite melt with the stoichiometry, NaCa$_2$Fe$_4$Mg$_{30}$Al$_3$Si$_{24}$O$_{89}$ to represent the Bulk Silicate Earth[15]. The oxide weight percent are reported in Table 1. We added 4CO (3.35 wt% CO), 8CO (5.16 wt% CO), 4CO$_2$ (6.48 wt% CO$_2$), and 8CO$_2$ (9.82 wt% CO$_2$) formula units to the 153-atom pyrolite supercell (see Table 2). Simulations were conducted at three temperatures (3000, 4000, and 5000 K) for pyrolite with the lower carbon concentrations (i.e., 3.35 wt% CO and 5.16 wt% CO$_2$) and at 4000 K for the carbon-rich compositions (i.e., 6.48 wt% CO and 9.82 wt% CO$_2$).

**Table 1 Initial pyrolite composition used in this study to which CO and $CO_2$ species were added**

|  | $SiO_2$ | $Al_2O_3$ | FeO | MgO | CaO | $Na_2O$ |
|---|---|---|---|---|---|---|
| This study | 44.5 | 4.72 | 9.00 | 37.3 | 3.46 | 0.96 |
| Pyrolite model | 45.0 | 4.45 | 8.05 | 37.8 | 3.55 | 0.36 |

A pyrolite model composition is shown for comparison[15]. The pyrolite model also includes 0.201 wt% $TiO_2$, 0.384 wt% $Cr_2O_3$, 0.135 wt% MnO, 0.25 wt% NiO, 0.029 wt% $K_2O$, and 0.021 wt% $P_2O_5$, which were not included in our simulations. Quantities are oxide weight percent

**Table 2 Carbon species added to pyrolite melt**

| Name | $CO_x$ per supercell | wt% $CO_x$ | wt% C | mol% C |
|---|---|---|---|---|
| "pyr4CO" | 4CO | 3.35 | 1.44 | 2.48 |
| "pyr4$CO_2$" | 4$CO_2$ | 5.16 | 1.41 | 2.42 |
| "pyr8CO" | 8CO | 6.48 | 2.78 | 4.97 |
| "pyr8$CO_2$" | 8$CO_2$ | 9.82 | 2.68 | 4.52 |

The four compositions examined in this study, expressed as the number of formula units of $CO_x$ added to a pyrolite supercell with 153 atoms, weight percent carbon monoxide or dioxide (wt% $CO_x$), weight percent carbon (wt% C), and mole percent carbon (mol% C)

**Code availability**. The Vienna Ab Initio Simulation Package (VASP) is a proprietary software available for purchase at https://www.vasp.at/. Data processing scripts written to process output files and create figures are available upon request.

## Data availability
The simulation output files are available upon request. They are not publicly available due to the very large file sizes. Parameters of the input files are described in computational methods.

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

## Acknowledgements
The authors would like to thank Z. Li, F. Soubiran, and A. Kobsch for valuable discussions. This research was supported in part by the European Research Council (ERC) under the European Union's Horizon 2020 research and innovation program (Grant agreement no. 681818—IMPACT to R.C.) and by the Deep Carbon Observatory. The authors acknowledge access to the GENCI supercomputers (Occigen, Ada, and Curie) through the stl2816 series of eDARI computing grants.

## Author contributions

R.C. designed the study and performed the calculations. N.V.S. performed a portion of the calculations, carried out data analysis, and wrote the manuscript. R.C., N.V.S., and C. E.M. interpreted the results, discussed the implications, and contributed to writing the manuscript.

## Additional information

**Competing interests:** The authors declare no competing interests.

