## [Peer Review File · Nature Communications]

Reviewers' comments:

Reviewer #1 (Remarks to the Author):

Page 2:

I think this is an interesting study with unique results but that the manuscript fails to live up to the promise established in the abstract. That is, not enough time is spent connecting the results to the implications or even expanding on these implications in the body of the text. Additionally, the results themselves are conveyed in a sort of a clumsy way, which reads a bit like someone taking dictation during a talk. While there is merit to a linear approach wherein the reader is walked through the plots one by one, small edits would go far to make this a smoother read. Despite these relatively minor complaints, I think after revisions this manuscript merits publication. If not in Nature Communications, in a journal which is more suited to its scope.

Suggested edits for the main text:

Page 2

"We find that with increasing pressure, the abundance of unpolymerized carbon species (CO₂, CO₃ and CO₄) decreases ..."

=> Looking at Figure 1a it does not seem like CO₄ decreases with pressure. I understand why that you are showing the % CO_x, but if this is a major point you should have a figure that shows this. Perhaps extending the compositions shown in Figure 2? Or simply adding a plot that shows polymerized versus CO_x as a function of pressure in different compositions?

The sentence "about half of carbon is polymerized at 100 GPa in melts..." is phrased ambiguously.

The phrase "the forms of carbon" is awkward. Perhaps replace with "carbon speciation".

Page 3

"Which may change the local structure and coordination environment of the atoms" => this merits a citation

I would amend the phrase "calculations allow" to specify "ab initio calculations".

"In low-pressure silicate melts, carbon behaves as a network modifier while at high pressure, it is thought that carbon will replace silicon and behave as a network former " => This sentence just needs to be ironed out for clarity.

"A detailed computational study can provide insight into carbon chemistry at high pressures and temperatures without the limitations of experimental techniques. "

=> I would replace with something like, "The detailed computational study presented here provides insights into high-pressure, high-temperature carbon chemistry without the limitations inherent to experiments conducted at extreme conditions."

"Density difference between carbon-bearing pyrolite melt and carbon-free pyrolite melt decreases with increasing pressure, plateauing above 10-20 GPa "

=> I think 'plateau' is an overstatement. It is clear that different temperatures/compositions have different trends, and that for many of them the difference gradually increases again after a local minima at ~30 GPa.

"To discern the behavior of oxidized carbon in the magma ocean with $x > 1$ we examine the relative proportions " => This is not clear and you should state "oxidized carbon (CO_x)".

Page 4

In regards to the first sentence, since Supplementary Figure 1 doesn't show the agreement with previous studies, you should move that to the next sentence: "plateauing above 10-20 GPa (Supplementary Fig. 1)".

Furthermore, I'm not sure that it is accurate to say that the compressibility is consistent with previous work since there are significant differences in the EoS parameters between these studies and yours. However, it does look like the addition of carbon has a similar influence on the compressibility of silicates in your calculations and previous experiments. I suggest rewording to clarify.

Regarding the first sentence under the 'carbon coordinated by oxygen' heading, it would be helpful to reintroduce 'C_xO_y' in this sentence before using the term ' $x > 1$ '.

"Between 0 and 10 GPa ... from about 40 and 60% ..." => I don't see these numbers reflected in the plots. It looks more like 40 and 50%.

And although there is a decrease to ~20% in pyr4CO₂ at what looks like a pressure <10 GPa, there seems to be a rebound to ~25% at least just above 10 GPa.

Where you say, "decrease to 0-15% at high pressures", you should clarify what you mean by high pressures. These two systems behave rather differently, particularly with the uptick in CO₂ in pyr4CO₂ at 60 GPa. Perhaps this section could be rewritten to maintain clarity but more accurately reflect these differences.

To the sentence "there is a sharp increase..." You should start out by clarifying, "in both the pyr4CO₂ and pyr8CO₂ systems, there is a sharp increase..."

However, the next sentence regarding the sharp increase in CO₄ only seems to reflect the pyr4CO₂ system, since it looks like the concentration in pyr8CO₂ approaches 80%. Because it is not clear which systems are being included in each sentence it is hard to draw conclusions.

Page 5

"The addition of more reduced carbon species results in higher concentration... at all pressures" => Figure S3 seems to indicate several crossover points at which pyr8CO has higher concentrations of CO₃/CO₄ than pyr4CO (e.g., CO₄ levels at P<80 GPa). If you didn't mean that there was an inverse relationship between the addition of reduced carbon species and a lower concentration of CO₃/CO₄, the sentence needs reworking.

End of page 5: you should move the definition of polymerization to the beginning of the paper when it was first introduced.

Page 6

It doesn't look like the plateau in pyr8CO₂ at 20% polymerization continues at high pressure. You need to adjust the X-axis so all available data are shown.

Page 9

Since your own results indicate a difference in the compressive properties of C-bearing and C-free

systems, I think you need to at least posit an alternate explanation for why this is so.

Figures

Figure 1:

Why does part a extend to higher pressures than parts b and c? Presumably, you have done the calculations, and you should extend plots b and c accordingly.

Add tick marks on X-axis of parts a, b, and c. If you extend plots b and c you could get rid of the X-axis label under part a.

Lastly, 1b, in particular, is hard to read because of the similarity between colors. The addition of more colors would make this plot more legible, as would using two shades of each color which are more distinct.

You also don't explain in the caption what the meaning of the shaded region is in 1b or 1c.

Figure 2:

Please move the key showing which color is 65 GPa versus 115 GPa to the top panel to make the plot more intuitive (since people read from the top down)

Add Figure?

It might be helpful to add a figure like Fig. 2, but instead of showing c=polymerization as a function of composition as a function of pressure (show 2-4 pressure steps).

Supplemental Materials

EoS: just include the equation

Figure 1a:

I don't see the solid light grey curve that's supposed to correspond to the carbon-free pyrolite. Also, since you are showing different compositions AND different temperatures, it would be helpful if you use different symbols for different compositions. I would also then use those in 1c.

It is clear that there is data for the pyrolite + 8CO/8CO₂ extends up to ~140 GPa yet you cut that data off here. You should plot this data even if it means rescaling all of the plots.

Caption: carbon-bearing pyrolite melt and carbon-free pyrolite melt instead of "dry"

Table 1: you need to define "exp" as experimental somewhere in the caption

Figure S2: It is hard to see the light grey color that indicates carbon or to distinguish between the two blue colors. It would be nice if the lines (dotted/dashed/solid) meant the same thing between S2a and S2c instead of meaning temperature in 2a and composition in 2c.

Figure S3: Consider changing the lines rather than just the symbols, as it is difficult to see which is which.

Figure S5. No explanation for the shaded region. I highly suggest removing it.

Reviewer #2 (Remarks to the Author):

I like this paper. I must say that I started reading it expecting much less; after all, ab initio MD has often been used with little creativity or even misused. Many questions and ready criticisms appeared immediately after reading the Abstract. However, when reading the manuscript all these questions were answered and I saw an elegant and convincing study, and a very well written manuscript. So my verdict is clearly "accept".

I could only add is about the oxidized vs reduced state of the melt. Here i have several points:

1. CO is a very unstable molecule, at all conditions.
2. It is clear that pyrolite rock is highly reducing, because of crystal chemistry of (Mg,Fe)SiO₃ perovskite, which demands that majority of Fe be Fe³⁺ and this can only happen in a disproportionation reaction $3\text{Fe}^{2+} \rightarrow 2\text{Fe}^{3+} + \text{Fe}^0$, releasing free metallic iron. The latter serves as a buffer, making the lower mantle extremely reducing.

In the pyrolite melt there should be no such preference for Fe³⁺, and no free metallic iron, and consequently, pyrolite melt in a sufficiently deep magma ocean or in the fully molten Earth does not have to be reducing. The authors could perhaps comment on the presence of Fe-clusters (analogs of free metallic Fe in pyrolite rock) in their melt (I expect none).

The good thing is that the main conclusions of this work don't really depend much on whether we take oxidized or reduced melt. The authors were careful enough to check and document this.

Response to Reviewers

Reviewer #1 (Remarks to the Author):

I think this is an interesting study with unique results but that the manuscript fails to live up to the promise established in the abstract. That is, not enough time is spent connecting the results to the implications or even expanding on these implications in the body of the text. Additionally, the results themselves are conveyed in a sort of a clumsy way, which reads a bit like someone taking dictation during a talk. While there is merit to a linear approach wherein the reader is walked through the plots one by one, small edits would go far to make this a smoother read. Despite these relatively minor complaints, I think after revisions this manuscript merits publication. If not in Nature Communications, in a journal which is more suited to its scope. ***We appreciate your valuable and detailed feedback. We have expanded the implications section and adjusted the writing of the manuscript to make it flow smoother. Thank you for recommending our manuscript for publication. We believe our manuscript would be an excellent contribution to Nature Communications, as our study fills a crucial void in our understanding of carbon in high-pressure high-temperature silicate melts and mechanisms for carbon sequestration to the core.***

From above: "That is, not enough time is spent connecting the results to the implications or even expanding on these implications in the body of the text."

We have added several sentences to the section "Carbon complexes with Fe and Si," describing in more detail the implications for core formation, along with additional references to support our implications.

From above: "Additionally, the results themselves are conveyed in a sort of a clumsy way, which reads a bit like someone taking dictation during a talk. While there is merit to a linear approach wherein the reader is walked through the plots one by one, small edits would go far to make this a smoother read."

Although we retain a linear approach for organizational purposes, we made edits to make the paper read smoother. We added qualifying sentences after quantifying statements and concluding statements at the end of paragraphs, which achieves smoother transitions and expands on the implications.

Suggested edits for the main text:

"We find that with increasing pressure, the abundance of unpolymerized carbon species (CO_2 , CO_3 and CO_4) decreases ..."

=> Looking at Figure 1a it does not seem like CO_4 decreases with pressure. I understand why that you are showing the % CO_x , but if this is a major point you should have a figure that shows this. Perhaps extending the compositions shown in Figure 2?

Thank you for your feedback. That is an accurate observation, and we have fixed this sentence in the abstract to read "the abundance of CO_2 and CO_3 species decreases at the expense of complex oxo-carbon polymers (C_xO_y) displaying multiple C-C bonds."

Or simply adding a plot that shows polymerized versus CO_x as a function of pressure in different compositions?

Figure 1b essentially shows this. The inverse of Figure 1b is the percent of carbon that is not polymerized (i.e., as CO_x). We have also added % C_xO_y species to Figure S2 to show that the abundances of CO₄ and C_xO_y both increase with increasing pressure.

The sentence “about half of carbon is polymerized at 100 GPa in melts...” is phrased ambiguously.

We have changed the sentence to read, "At 100 GPa, about half of carbon atoms are polymerized (i.e., bonded to carbon) in pyrolite melts..."

The phrase “the forms of carbon” is awkward. Perhaps replace with “carbon speciation”.

We have changed "the forms of carbon" to "the speciation of carbon" in all instances.

Page 3

“Which may change the local structure and coordination environment of the atoms” => this merits a citation

A citation has been added.

I would amend the phrase “calculations allow” to specify “ab initio calculations”.

We have added this specification.

“In low-pressure silicate melts, carbon behaves as a network modifier while at high pressure, it is thought that carbon will replace silicon and behave as a network former “ => This sentence just needs to be ironed out for clarity.

We have changed the sentence to read, "At ambient pressure, carbon behaves as a network modifier (i.e., exists as CO_x species that break up the silicate network) while at high pressure, it is thought that carbon will behave as a network former by replacing silicon."

“A detailed computational study can provide insight into carbon chemistry at high pressures and temperatures without the limitations of experimental techniques. “

=> I would replace with something like, “The detailed computational study presented here provides insights into high-pressure, high-temperature carbon chemistry without the limitations inherent to experiments conducted at extreme conditions.”

We have changed the sentence to your suggestion.

“Density difference between carbon-bearing pyrolite melt and carbon-free pyrolite melt decreases with increasing pressure, plateauing above 10-20 GPa “

=> I think 'plateau' is an overstatement. It is clear that different temperatures/compositions have different trends, and that for many of them the difference gradually increases again after a local minima at ~30 GPa.

We have changed the sentence to read, "reaching a local minimum at ~30 GPa after which the density difference between the melts increases again."

“To discern the behavior of oxidized carbon in the magma ocean with $x > 1$ we examine the relative proportions “ => This is not clear and you should state “oxidized carbon (CO_x)”.”
We have changed the sentence to read, "To discern the behavior of oxidized carbon in the magma ocean, we examine the relative proportions of CO_2 , CO_3 and CO_4 species..."

Page 4

In regards to the first sentence, since Supplementary Figure 1 doesn't show the agreement with previous studies, you should move that to the next sentence: “plateauing above 10-20 GPa (Supplementary Fig. 1)”.

We have moved "Supplementary Fig. 1" as a reference to the next sentence.

Furthermore, I'm not sure that it is accurate to say that the compressibility is consistent with previous work since there are significant differences in the EoS parameters between these studies and yours. However, it does look like the addition of carbon has a similar influence on the compressibility of silicates in your calculations and previous experiments. I suggest rewording to clarify.

We have changed it to read, "...the effect of carbon on the melt's compressibility behavior is consistent with previous work..."

Regarding the first sentence under the ‘carbon coordinated by oxygen’ heading, it would be helpful to reintroduce ‘ C_xO_y ’ in this sentence before using the term ‘ $x > 1$ ’.

We have simply removed " $x > 1$ " to avoid confusion.

“Between 0 and 10 GPa ... from about 40 and 60% ...” => I don't see these numbers reflected in the plots. It looks more like 40 and 50%.

We have changed "40 and 60%" to read "40 and 50%."

And although there is a decrease to ~20% in pyr_4CO_2 at what looks like a pressure <10 GPa, there seems to be a rebound to ~25% at least just above 10 GPa.

This is likely due to scatter in the data, specifically in the quantity of CO_2 .

Where you say, “decrease to 0-15% at high pressures”, you should clarify what you mean by high pressures. These two systems behave rather differently, particularly with the uptick in CO_2 in pyr_4CO_2 at 60 GPa. Perhaps this section could be rewritten to maintain clarity but more accurately reflect these differences.

We have clarified it with, "followed by a more gradual decrease in concentration to 0-15% at lowermost mantle pressures"

To the sentence “there is a sharp increase...” You should start out by clarifying, “in both the pyr_4CO_2 and pyr_8CO_2 systems, there is a sharp increase...”

We have changed the sentence to read, "In both pyr_4CO_2 and pyr_8CO_2 systems, there is a sharp increase..."

However, the next sentence regarding the sharp increase in CO₄ only seems to reflect the pyr4CO₂ system, since it looks like the concentration in pyr8CO₂ approaches 80%. Because it is not clear which systems are being included in each sentence it is hard to draw conclusions. ***We changed the sentence to read, "reaching 60% at 100 GPa for both pyr4CO₂ and pyr8CO₂ systems." The CO₄ concentration in pyr8CO₂ is indeed 80% at ~135 GPa, but as we don't have data for pyr4CO₂ at the equivalent pressure, we make the comparison at 100 GPa.***

Page 5

"The addition of more reduced carbon species results in higher concentration... at all pressures" => Figure S3 seems to indicate several crossover points at which pyr8CO has higher concentrations of CO₃/CO₄ than pyr4CO (e.g., CO₄ levels at P<80 GPa). If you didn't mean that there was an inverse relationship between the addition of reduced carbon species and a lower concentration of CO₃/CO₄, the sentence needs reworking.

We meant pyr4CO and pyr8CO relative to pyr4CO₂ and pyr8CO₂. We have changed the sentence to read, "Compared to the more oxidized melts (pyr4CO₂ and pyr8CO₂), the more reduced melts (pyr4CO and pyr8CO) have a higher concentration of CO₂ and a lower concentration of CO₃ and CO₄ at all pressures."

End of page 5: you should move the definition of polymerization to the beginning of the paper when it was first introduced.

We have added a definition to the introductory paragraph: "mechanism by which carbon polymerizes (where carbon atoms bond to other carbon atoms)."

Page 6

It doesn't look like the plateau in pyr8CO₂ at 20% polymerization continues at high pressure. You need to adjust the X-axis so all available data are shown.

The x-axis has been adjusted. We changed the sentence to read "...pyr8CO₂ experiences a shallow increase in polymerization to about 30% at maximum pressure."

Page 9

Since your own results indicate a difference in the compressive properties of C-bearing and C-free systems, I think you need to at least posit an alternate explanation for why this is so. ***We have added a discussion of this in Supplementary Materials in the Equation of State section. It likely due to the fact that we are adding carbon in the form of CO and CO_x, which also means that we add oxygen to the system. The addition of oxygen, which is highly compressible relative to cations, is likely responsible for the lower bulk modulus of the carbonated melts.***

Figures

Figure 1:

Why does part a extend to higher pressures than parts b and c? Presumably, you have done the calculations, and you should extend plots b and c accordingly.

The x-axes have been extended to 140 GPa.

Add tick marks on X-axis of parts a, b, and c. If you extend plots b and c you could get rid of the X-axis label under part a.

Tick marks have been added. The x-axis label under part a has been removed.

Lastly, 1b, in particular, is hard to read because of the similarity between colors. The addition of more colors would make this plot more legible, as would using two shades of each color which are more distinct.

The colors in fact aren't needed as the line style and marker shapes are different. We made the colors more distinct from each other. We agree that it improves the figure. We tried using different colors (e.g., green, blue, orange), but it proved to be more distracting than useful.

You also don't explain in the caption what the meaning of the shaded region is in 1b or 1c.

We have added explanations, as follow, "1b: Shaded regions represent the range from the more oxidized to the more reduced melts for each temperature and/or carbon content. and for 1c: Shaded region represents the approximate region that pyrolite + 4CO encompasses to highlight the contrast with pyrolite + 8CO."

Figure 2:

Please move the key showing which color is 65 GPa versus 115 GPa to the top panel to make the plot more intuitive (since people read from the top down)

Fixed. We moved the legend to the top panel.

Add Figure?

It might be helpful to add a figure like Fig. 2, but instead of showing c=polymerization as a function of composition as a function of pressure (show 2-4 pressure steps).

Fig. 2 is a function of composition and pressure. We show two pressures (intermediate pressure at around 65 GPa and high pressure at around 115 GPa). 0 GPa would show no polymerized species. Our pressure steps are not small or equivalent enough (between compositions) to show more pressure steps effectively.

Supplemental Materials

EoS: just include the equation

The fourth-order Birch-Murnaghan equation has been added.

Figure 1a:

I don't see the solid light grey curve that's supposed to correspond to the carbon-free pyrolite.

Removed from the caption.

Also, since you are showing different compositions AND different temperatures, it would be helpful if you use different symbols for different compositions. I would also then use those in 1c. ***Fixed. The reduced compositions are now represented with diamond symbols and the oxidized compositions are circles in both 1a and 1c.***

It is clear that there is data for the pyrolite + 8CO/8CO₂ extends up to ~140 GPa yet you cut that data off here. You should plot this data even if it means rescaling all of the plots.

All plots have been rescaled to 140 GPa.

Caption: carbon-bearing pyrolite melt and carbon-free pyrolite melt instead of "dry"

Fixed.

Table 1: you need to define "exp" as experimental somewhere in the caption

Added the definition to the caption.

Figure S2: It is hard to see the light grey color that indicates carbon or to distinguish between the two blue colors.

The concentration of elemental C is 0 at all pressures, so we simply removed it. We split up the figure into several panels and changed the symbols for each species. We also added the species "CxOy" to reflect your previous observation that the concentration of both CO₄ and CxOy species increase with increasing pressure.

It would be nice if the lines (dotted/dashed/solid) meant the same thing between S2a and S2c instead of meaning temperature in 2a and composition in 2c.

Fixed. The symbols are now different shapes and colors for each species, and the symbol/color/line representations are consistent across all panels.

Figure S3: Consider changing the lines rather than just the symbols, as it is difficult to see which is which.

We removed this figure as it is redundant with the modified Figure S2.

Figure S5. No explanation for the shaded region. I highly suggest removing it.

Removed.

Reviewer #2 (Remarks to the Author):

I like this paper. I must say that I started reading it expecting much less; after all, ab initio MD has often been used with little creativity or even misused. Many questions and ready criticisms appeared immediately after reading the Abstract. However, when reading the manuscript all these questions were answered and I saw an elegant and convincing study, and a very well written manuscript. So my verdict is clearly “accept”.

Thank you very much for your recommendation for the publication of our manuscript.

I could only add is about the oxidized vs reduced state of the melt. Here i have several points:

1. CO is a very unstable molecule, at all conditions.

Indeed. Please find a revised Supplementary Figure 2 which shows all carbon species for all of our melt compositions at all investigated temperatures, as a function of pressure. Above 20 GPa, CO concentrations fall below about 10% for all compositions.

2. It is clear that pyrolite rock is highly reducing, because of crystal chemistry of (Mg,Fe)SiO₃ perovskite, which demands that majority of Fe be Fe³⁺ and this can only happen in a disproportionation reaction $3\text{Fe}^{2+} \rightarrow 2\text{Fe}^{3+} + \text{Fe}^0$, releasing free metallic iron. The latter serves as a buffer, making the lower mantle extremely reducing. In the pyrolite melt there should be no such preference for Fe³⁺, and no free metallic iron, and consequently, pyrolite melt in a sufficiently deep magma ocean or in the fully molten Earth does not have to be reducing.

Indeed, a pyrolite composition is highly reducing, but with the additions of CO and CO₂, we increase the oxidation state. Although there is much uncertainty on the oxidation state of Earth's magma ocean, there is a general consensus that the oxidation state of Early Earth's mantle (i.e., prior to the mantle great oxidation event) was significantly more reduced (well below the iron-wustite buffer) compared to present-day Earth (Wade and Wood, 2005; Scaillet and Gaillard, Nature 2011; Gaillard et al., Chemical Geology 2015). We added the sentence, "It is thought that the oxidation state of Early Earth's mantle was far below the iron-wustite buffer prior to core-mantle separation, subsequently becoming more oxidized due to convective mixing" to the introductory paragraph with relevant citations.

The authors could perhaps comment on the presence of Fe-clusters (analogs of free metallic Fe in pyrolite rock) in their melt (I expect none).

We do not observe direct Fe-Fe bonding, but instead direct Fe-C bonds and clustering of iron polyhedra (e.g., FeO₅, FeO₆) (see Fig. 3). Our silicate melt is not so reduced to form iron metal, nor can we observe phase separation as we are simulating one phase, pyrolite melt.

The good thing is that the main conclusions of this work don't really depend much on whether we take oxidized or reduced melt. The authors were careful enough to check and document this.

Indeed, we demonstrate the presence of C-C, Fe-C and Si-C clustering in all of our melts, noting that the abundance is increased for a more reduced melt compared to a more oxidized melt.